# Evaluating supply chain management of SARS-CoV-2 point-of-care (POC) diagnostic services in primary healthcare clinics in Mopani District, Limpopo Province, South Africa

**Kuhlula Maluleke** [1]*, **Alfred Musekiwa**[1], **Tivani Mashamba-Thompson**[2]

**1** School of Health Systems and Public Health, Faculty of Health Sciences, University of Pretoria, Pretoria, South Africa, **2** Faculty of Health Sciences, University of Pretoria, Pretoria, South Africa

* u15266304@tuks.co.za

## Abstract

Access to point-of-care (POC) diagnostics in resource-limited settings, where laboratory-based diagnostics are limited, depends on efficient supply chain management (SCM). This study evaluated the SCM for SARS-CoV-2 POC diagnostic services in resource-limited settings to determine the effect of SCM on accessibility to SARS-CoV-2 POC tests and to identify barriers and enablers of accessibility to SARS-CoV-2 diagnostic services in Mopani District, Limpopo Province, South Africa. We purposively assessed 47 clinics providing POC diagnostic services between June and September 2022. One participant per clinic completed an audit tool developed by the authors with guidance from the World Health Organization and the Management Sciences for Health guidelines. The audit tool evaluated the following SCM parameters: selection, quantification, storage, procurement, quality assurance, distribution, redistribution, inventory management, and human resource capacity. Percentage rating scores between 90–100% indicated that the facility was compliant with SCM guidelines, while rating scores < 90% indicated non-compliance. The clinic audit scores were summarized and compared across clinics and sub-districts. Clinics had compliance scores ranging from 60.5% to 89.2%. Compliance scores were the highest for procurement, redistribution, and quality assurance (all 100%), followed by storage (mean = 95.2%, 95% CI: 90.7–99.7), quantification (mean = 89.4%, 95% CI: 80.2–98.5), and selection (mean = 87.5%, 95% CI: 87.5%–87.5%). Compliance scores were the lowest for inventory management (mean = 53.2%, 95% CI: 47.9%–58.5%), distribution (mean = 48.6%, 95% CI: 44.6%–52.7%), and human resource capacity (mean = 50.6%, 95% CI: 43.3%–58.0%). A significant correlation was found between compliance score and clinic headcount (r = 0.4, p = 0.008), and compliance score and ideal clinic score (r = 0.4, p = 0.0003). Overall, the 47 clinics audited did not comply with international SCM guidelines. Of the nine SCM parameters evaluated, only procurement, redistribution, and quality assurance did not need improvement. All parameters are key in ensuring full

**Data Availability Statement:** All relevant data are within the paper and its Supporting information files.

**Funding:** The author(s) received no specific funding for this work.

**Competing interests:** The authors have declared that no competing interests exist.

**Abbreviations:** CI, Confidence Interval; COVID-19, Coronavirus 19; CSA, Covid Screening Application; DoH, Department of Health; EDL, Essential Drug List; LMIC, Low- and middle-income countries; MSH, Management Sciences for Health; NDoH, National Department of Health; NHLS, National Health Laboratory Service; OPM, Operational Manager; OPM, Operational Manager; PHC, Primary Healthcare; PI, Principal Investigator; POC, Point-of-care; SARS-CoV-2, Severe acute respiratory syndrome coronavirus type 2; SCM, Supply chain management; SVS, Stock Visibility System; WHO, World Health Organisation.

functionality of SCM systems and equitable access to SARS-CoV-2 POC diagnostics in resource limited settings.

## Introduction

Severe acute respiratory syndrome coronavirus 2 (SARS-CoV-2), the virus that causes coronavirus disease 2019 (COVID-19), cannot be properly identified or managed without correct diagnosis [1]. Timely access to health services enables rapid testing and prevents disease progression, resulting in improved individual and public health outcomes [2, 3]. Access to health services is defined as the ability to use services when and where they are needed [4]. In resource-limited settings where access to laboratories is limited, primary healthcare (PHC) is the center point of access to health services, including access to point-of-care (POC) diagnostic services. The introduction of POC tests in resource-limited settings have been proven to be effective for strengthening health systems by providing rapid results to improve timely initiation of suitable therapy, facilitate linkages to care, and improve health outcomes [5].

Supply chain management (SCM) includes the resources and processes needed to deliver the goods and services for POC diagnostic services [6]. Poor SCM systems may lead to stock outs of POC diagnostic tests [7]. This causes a ripple effect because stock outs result in the reduced use of POC diagnostic tests, which negatively impacts health outcomes [8–10].

The procurement of essential drugs and health-related commodities by the South African government follows a closed tender system, with drug distribution limited to the Essential Drug List and registered products [11]. The National Department of Health manages the tender system with input from the National Treasury, while the provinces procure their drugs from preferred suppliers on the national database [12–14]. The medications are then repacked at provincial depots before being sent to district hospitals and eventually to PHC facilities [13]. However, irregularities during the COVID-19 pandemic emergency procurement strategy were identified due to the lack of an adequate regulatory framework [12].

Evidence shows that SCM systems can be influenced by both barriers and enabling factors, including selection, quantification, storage, procurement, quality assurance, distribution, redistribution, inventory management, and human resource capacity [15]. As the demand for POC tests and other COVID-19 medical supplies increased in 2020 due to an increase in the number of daily cases, various SCM systems, including procurement, transportation, and manufacturing, were disrupted [16, 17]. During 2020 and through 2021, there were border closures, lockdowns in the supply market, interruptions in vehicle movements and international trade, labor shortages, and irregularities in health and safety protocols in manufacturing facilities [18].

As the COVID-19 pandemic draws to a close, POC testing has made great strides in diagnosing and managing the disease burden in Africa [19]. The results of this study will be useful to both supply chain managers and policymakers. Supply chain managers can use these findings to monitor and improve quality over time at individual, sub-district and district levels. Policy makers who are responsible for implementing POC diagnostic services in PHC clinics in resource-limited settings can identify strategies that could be embedded to the current policies to encourage successful implementation of POC diagnostic services.

The main objective of this study was to evaluate the SCM for SARS-CoV-2 POC diagnostic services in resource-limited settings. This evaluation was aimed at determining the effect of SCM on accessibility to SARS-CoV-2 POC tests and to identify barriers and enablers of accessibility to SARS-CoV-2 diagnostic services.

## Methods

### Study design

This study is the third phase of a multi-phase study, which aimed to develop a novel approach for improving SCM for SARS-CoV-2 POC diagnostic services in resource-limited settings. We used the Mopani District in Limpopo Province, South Africa as a case study [20]. In this phase, we conducted an audit of PHC clinics in the Mopani District using a cross-sectional survey designed to collect and collate data relating to the prevalence of particular events [21]. We compared the collected benchmark data to a set of well-defined standards that were previously used to identify the changes needed to improve the quality of POC diagnostic services [21, 22].

### Study population

This study was conducted in the Mopani District, one of the five districts in Limpopo Province, South Africa. The district has five sub-districts: Ba-phalaborwa, Greater Giyani, Maruleng, Greater Tzaneen, and Greater Letaba (Fig 1). This resource-limited setting was selected

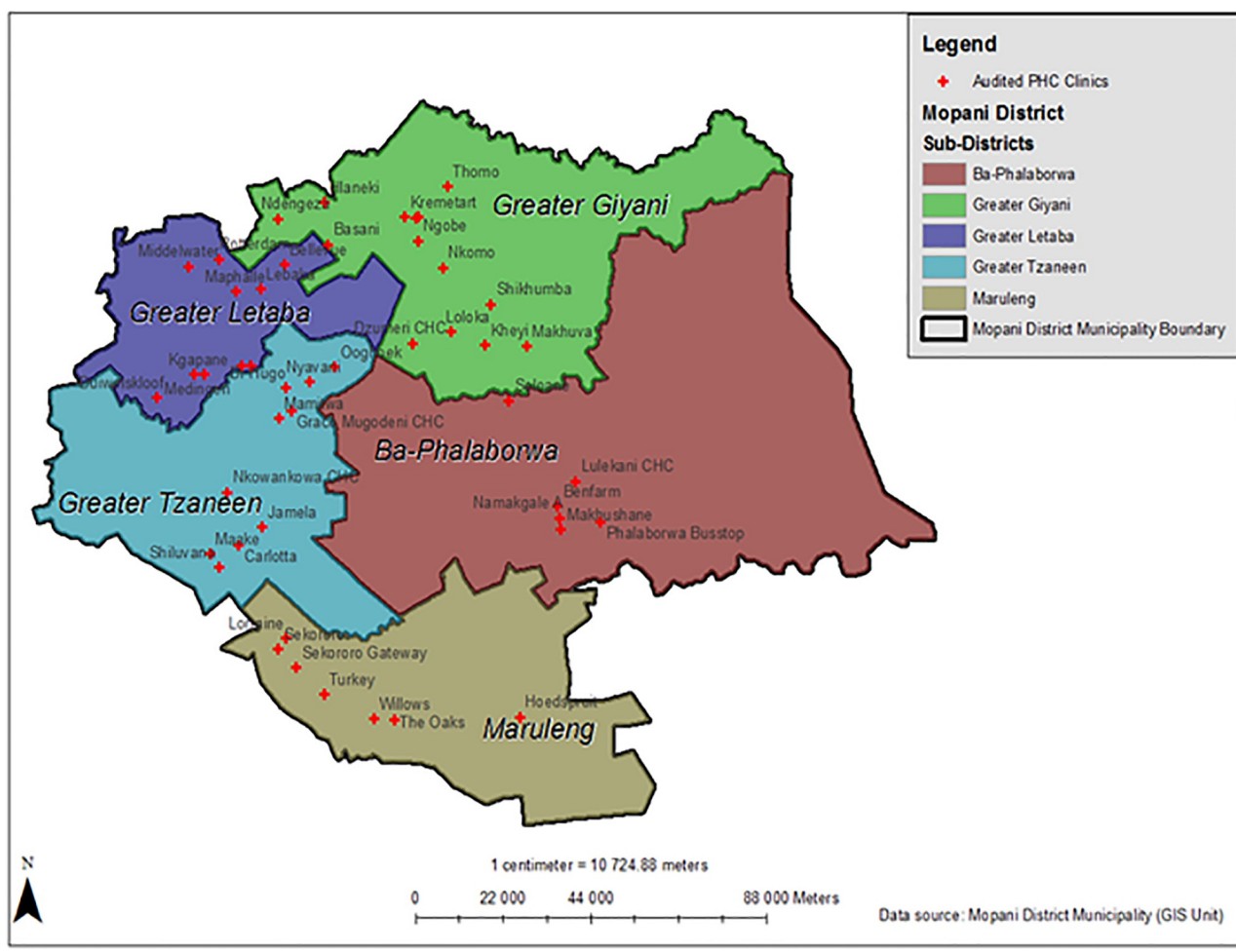

**Fig 1. Geographic location and distribution of the audited primary healthcare clinics in the Mopani District, Limpopo province, South Africa.** Image created by authors on ESRI's ArcGIS 10.8.2; no copyrighted material was used.

because 81% of the population reside in rural areas, 14% in urban areas, and 5% on farms [23]. Most rural residents are poor and have no income because the local economy cannot provide remunerative jobs or self-employment opportunities [24]. As a result, most of the population in the study area relied on the public health system for health services.

## Sampling strategy

We purposively sampled 47 out of 105 PHC clinics to participate in the study. This sampling technique involved selecting PHC clinics based on the size of the serviced population. PHC clinics with large (> 19 000), moderate (10 001–19 000) and small (3 000–10 000) populations were selected to ensure generalizability of the audit results.

## Recruitment strategy

We contacted PHC clinic sub-district managers to ask for access to the PHC clinics and request participation of clinic staff. Upon approval from the PHC clinic sub-district managers, we sent a request to the operational managers to visit the clinics. The personnel involved in the SCM system, operational managers, professional nurses, and pharmacist assistants were asked to complete the audit tool.

## Audit tool

The audit tool used (S1 Fig) in this study was developed based on the WHO guidelines for improving the quality of POC testing [25] and the Management Sciences for Health (MSH) diagnostic SCM guideline [26]. We focused solely on aspects related to SCM and used these documents to evaluate the challenges that resource-limited settings face in accessing appropriate, quality-assured, and adequate POC diagnostics, as well as to identify potential strategies for addressing these challenges. The audit tool included clinic characteristics such as: i) annual headcount, or number of patients who present to the PHC clinic regardless of the health service provided; and ii) the ideal clinic score, where an ideal clinic is defined as a clinic with good infrastructure, adequate staff, adequate medicine and supplies, good administrative processes, and sufficient adequate bulk supplies [27].

## Data collection

We conducted an audit of the sampled PHC clinics from June to September 2022. The audit team consisted of the principal investigator (PI) and a research assistant. The audit questionnaire was administered by the PI after presenting a summary of the study and obtaining consent from participants. The audit was carried out at the PHC clinics where the participants worked. To ensure that the audit did not affect the daily operations and duties at the PHC clinic, we adhered to the scheduled appointments and time limits to complete the questionnaire. We followed the national COVID-19 regulations, guidelines, and protocols. We ensured that we had a COVID-19 researcher toolkit when interacting with the participants. This toolkit included own mask, masks for participants, thermometer, alcohol-based hand sanitizer, sanitizer for surfaces, a box to put all informed consent forms in, and paper-based questions, box of tissues, and a bag for disposal of used masks and tissues.

## Scoring and rating guide

The audit tool had 42 questions categorized into 9 main sections, including selection, quantification, storage, procurement, quality assurance, distribution, re-distribution, inventory management, and human resource capacity. The response modalities of the questions were "yes",

"no" and "n/a". If "no" or "n/a" responses were selected, further explanation was probed for under the comment section to establish the barriers and enablers of SCM on accessibility of SARS-CoV-2 POC diagnostic services.

To evaluate how compliant PHC clinics were with SCM guidelines for SARS-CoV-2 POC diagnostic services, we summarized the audit scores. A point was allocated to each question when all the requirements were met. We summed all the rating scores for each component to obtain the percentage rating score per component (S1 Table). The overall percentage rating score was the sum of all the rating scores for each component. Compliance to SCM was interpreted as follows: rating scores between 90–100% indicated that the PHC clinic was compliant to SCM guidelines (satisfactory compliance), rating scores < 90% indicated non-compliance to SCM guidelines (unsatisfactory compliance).

## Statistical methods

The data were captured in a Microsoft Excel spreadsheet, which was then cleaned, validated, and exported to Stata software version 17 for analysis. Frequencies, means, standard deviations, as well as 95% confidence intervals (CIs) for the audit scores were calculated. An analysis of variance test was conducted to test any differences between the compliance scores of the sub-districts followed by a Bonferroni post-hoc test to confirm where the differences occurred between the sub-districts. A further pairwise correlation analysis was done to establish any relationship between the overall SCM compliance score and the clinic characteristics. The results were statistically significant at $p \leq 0.05$. A qualitative thematic analysis of the nine components of the SCM system was conducted on R software.

## Ethical considerations

Permission was obtained from Mopani District Directorate before conducting this study. Ethical approval of the main study was granted by the University of Pretoria, Faculty of Health Sciences, Research Ethics Committee (Reference No: 655/2021, Dated: November 24, 2021) and the Limpopo Department of Health Ethics Committee (Reference No: LP_2021-12-007, Dated: February 27, 2022). All study participants signed informed consent before participating in the study. Participant names were kept confidential. We de-identified the facilities to ensure that the compliance score could not be linked to a facility.

## Results

### Characteristics of the PHC clinics in Mopani

We audited 47 PHC clinics. Of these, 14 were in Greater Giyani, seven in Maruleng, six in Ba-Phalaborwa, 10 in Greater Letaba, and 10 in Greater Tzaneen. The total population of the audited PHC clinics ranged from 4,534 to 29,605. The clinic annual headcounts, the number of patients who present to the PHC clinic regardless of the health service provided, of the audited PHC clinics ranged from 13,431 to 78,036. The mean ideal clinic score was 71% (95% CI: 66.9%–75.1%). All the audited PHC clinics provided COVID-19 POC diagnostic services, specifically the Abbott Panbio™ COVID-19 Antigen rapid test. The audit tool was completed by 27 (57.45%) operational managers (OPMs), 19 (40.43%) professional nurses, and 1 (2.12%) pharmacist assistant. These professionals are part of the SCM of COVID-19 POC tests and they identified themselves as inventory controllers, procurement officers, and end-users. (Table 1).

**Table 1. Characteristics of audited PHC clinics in the Mopani District, Limpopo Province, South Africa.**

| Name of Clinic | Total Population | Headcount | Ideal clinic score (%) | SARS-CoV-2 point-of-care testing | Occupation of assessor |
|---|---|---|---|---|---|
| Clinic 1 | 15538 | 38671 | 76 | Yes | Professional nurse |
| Clinic 2 | 8431 | 22651 | 66 | Yes | OPM* |
| Clinic 3 | 7052 | 19906 | 61 | Yes | OPM* |
| Clinic 4 | 6731 | 18025 | 80 | Yes | Professional nurse |
| Clinic 5 | 20914 | 52697 | 65 | Yes | Professional nurse |
| Clinic 6 | 4534 | 13431 | 59 | Yes | Professional nurse |
| Clinic 7 | 6045 | 17305 | 88 | Yes | OPM* |
| Clinic 8 | 9467 | 27130 | 40 | Yes | OPM* |
| Clinic 9 | 29326 | 78036 | 62 | Yes | OPM* |
| Clinic 10 | 8513 | 22485 | 78 | Yes | Professional nurse |
| Clinic 11 | 17116 | 53813 | 39 | Yes | Professional nurse |
| Clinic 12 | 10218 | 28970 | 63 | Yes | OPM* |
| Clinic 13 | 9103 | 25277 | 64 | Yes | OPM* |
| Clinic 14 | 8528 | 21401 | 57 | Yes | Professional nurse |
| Clinic 15 | 29605 | 56461 | 93 | Yes | OPM* |
| Clinic 16 | 12274 | 23457 | 54 | Yes | OPM* |
| Clinic 17 | 15193 | 29921 | 70 | Yes | OPM* |
| Clinic 18 | 8514 | 17604 | 91 | Yes | Professional nurse |
| Clinic 19 | 14655 | 31554 | 83 | Yes | OPM* |
| Clinic 20 | 10947 | 22630 | 95 | Yes | OPM* |
| Clinic 21 | 6591 | 19777 | 79 | Yes | Professional nurse |
| Clinic 22 | 7784 | 20577 | 71 | Yes | OPM* |
| Clinic 23 | 6317 | 24338 | 60 | Yes | Professional nurse |
| Clinic 24 | 9201 | 21880 | 85 | Yes | OPM* |
| Clinic 25 | 7629 | 20746 | 66 | Yes | Pharmacist Assistant |
| Clinic 26 | 11377 | 37707 | 52 | Yes | Professional nurse |
| Clinic 27 | 10274 | 32896 | 61 | Yes | OPM* |
| Clinic 28 | 7145 | 18657 | 71 | Yes | OPM* |
| Clinic 29 | 7208 | 19827 | 78 | Yes | Professional nurse |
| Clinic 30 | 13722 | 32370 | 75 | Yes | Professional nurse |
| Clinic 31 | 10970 | 25811 | 79 | Yes | Professional nurse |
| Clinic 32 | 13102 | 31173 | 57 | Yes | OPM* |
| Clinic 33 | 8638 | 17436 | 51 | Yes | OPM* |
| Clinic 34 | 10955 | 27708 | 69 | Yes | Professional nurse |
| Clinic 35 | 10013 | 24592 | 44 | Yes | Professional nurse |
| Clinic 36 | 15862 | 36750 | 91 | Yes | OPM* |
| Clinic 37 | 11241 | 27520 | 84 | Yes | OPM* |
| Clinic 38 | 19398 | 34021 | 79 | Yes | Professional nurse |
| Clinic 39 | 12993 | 27533 | 87 | Yes | Professional nurse |
| Clinic 40 | 20400 | 42884 | 71 | Yes | Professional nurse |
| Clinic 41 | 11167 | 22738 | 72 | Yes | OPM* |
| Clinic 42 | 17615 | 42946 | 80 | Yes | OPM* |
| Clinic 43 | 20076 | 45065 | 88 | Yes | OPM* |
| Clinic 44 | 19226 | 41503 | 62 | Yes | OPM* |
| Clinic 45 | 17688 | 41582 | 84 | Yes | OPM* |
| Clinic 46 | 12419 | 27234 | 71 | Yes | OPM* |

*(Continued)*

**Table 1.** (Continued)

| Name of Clinic | Total Population | Headcount | Ideal clinic score (%) | SARS-CoV-2 point-of-care testing | Occupation of assessor |
|---|---|---|---|---|---|
| Clinic 47 | 8598 | 17079 | 84 | Yes | OPM* |

*OPM = Operational Manager

### Overall audit scores for the audited PHC clinics in Mopani District

The audit results showed that the average SCM compliance score ranged between 60.5% and 89.2%. Based on our criteria, none of the clinics had satisfactory SCM compliance scores, with all PHC clinics scoring < 90%. The highest score of 89.2% was recorded for Clinics 9, 38, 40, 43, 44, and 47 (Fig 2).

### Component scores per PHC clinic

The audit evaluated the nine components of the SCM system: selection, quantification, storage, procurement, quality assurance, distribution, redistribution, inventory management and human resource capacity. All the PHC clinics scored highest for procurement, redistribution, and quality assurance with mean rating scores of 100%, followed by storage (mean = 95.2%. 95% CI: 90.7–99.7), quantification (mean = 89.4%, 95% CI: 80.2–98.5), and selection (mean = 87.5%, 95% CI: 87.5%–87.5%). The lowest mean ratings were found for inventory management (mean = 53.2%, 95% CI: 47.9%–58.5%), distribution (mean = 48.6%, 95% CI: 44.6%–52.7%), and human resource capacity (mean = 50.6%, 95% CI: 43.3%–58.0%) (Fig 3).

### Audit component scores per sub-district

We analyzed the SCM compliance scores per sub-district. Greater Tzaneen obtained the highest audit score (mean = 86.4%, 95% CI: 84.2%–88.6%). This was followed by Ba-Phalaborwa (mean = 82.1%, 95% CI: 80.5%–83.7%), Maruleng (mean = 80.9%, 95% CI: 77.6%–84.3%), and Greater Letaba (mean = 80.3%, 95% CI: 77.4%–83.3%). Greater Giyani had the lowest compliance score (mean = 75.5%, 95% CI: 70.6%–80.5%). Greater Giyani also had the highest variation in audit scores with an interquartile range (IQR) of 28.7%. Ba-Phalaborwa had the least varied audit scores with an IQR of 3.8% (Fig 4).

Post-hoc analysis of the SCM components per sub-district revealed that Greater Tzaneen and Greater Giyani differed significantly in terms of overall score (p = 0.004), quantification (p = 0.026), distribution (p = 0.033), and human resource management (p = 0.0009). Greater Giyani and Greater Letaba differed significantly in terms of quantification (p = 0.026). In terms of human resource capacity, significant differences were observed between Greater Tzaneen and Ba-Phalaborwa (p = 0.046), and Greater Tzaneen and Greater Letaba (p = 0.005). There were no significant differences observed for selection, storage, inventory management, procurement, re-distribution, and quality assurance between the sub-districts.

### Barriers and enablers of the supply chain management audit components

We qualitatively described the barriers and enablers of each individual component below, while providing a summary of the SCM component scores in Fig 3.

**Selection.** The mean score for selection of POC tests was 87.5% (95% CI: 87.5%–87.5%). Selection comprised seven questions. All the PHC clinics noted that the staff was not actively involved in selecting POC tests. Participants reported that the POC diagnostic tests were sensitive with very few false negative and very few false positives. Participants reported that the

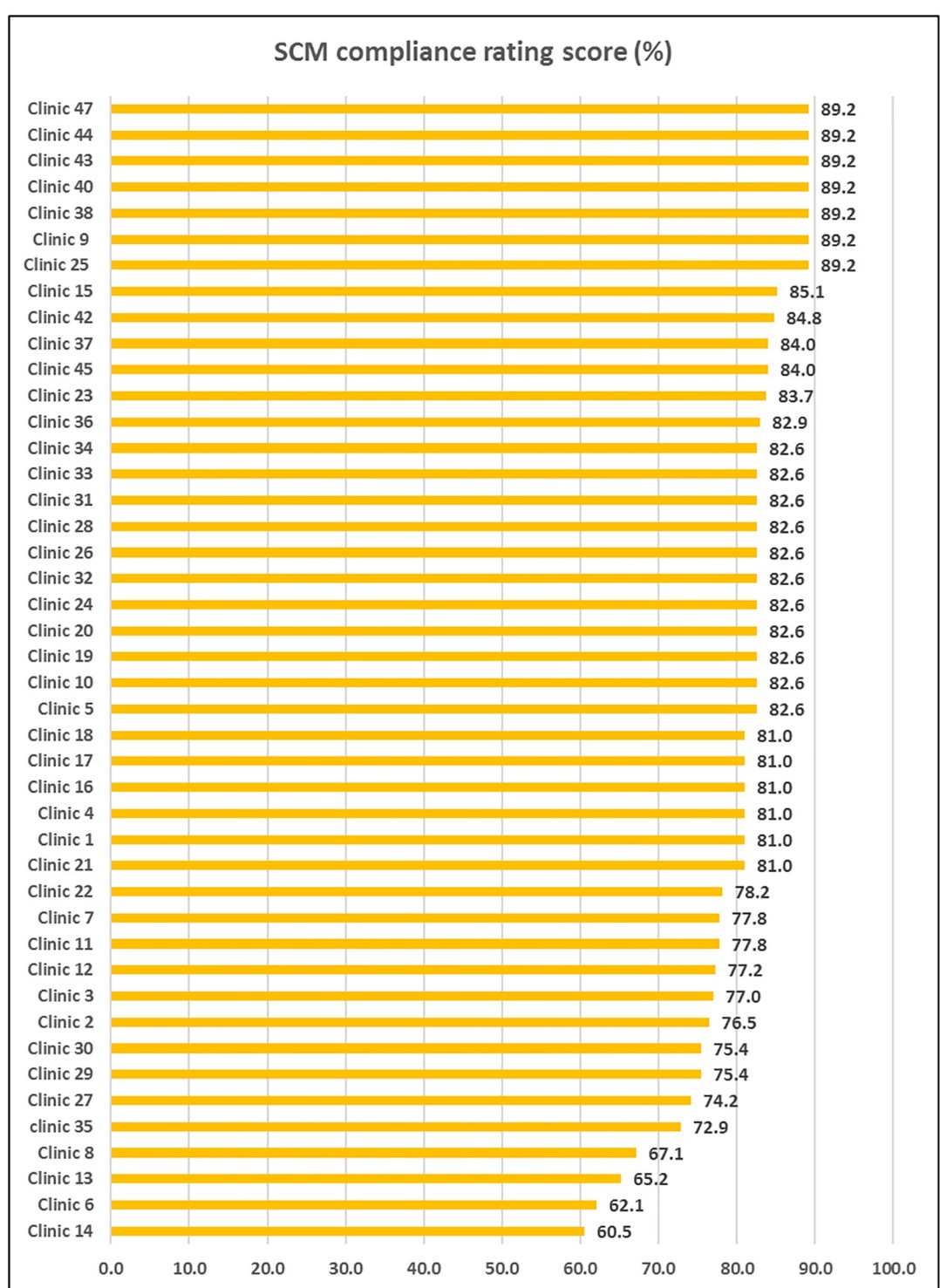

**Fig 2. Supply chain management compliance rating of the audited PHC clinics, Mopani District, Limpopo Province, South Africa.**

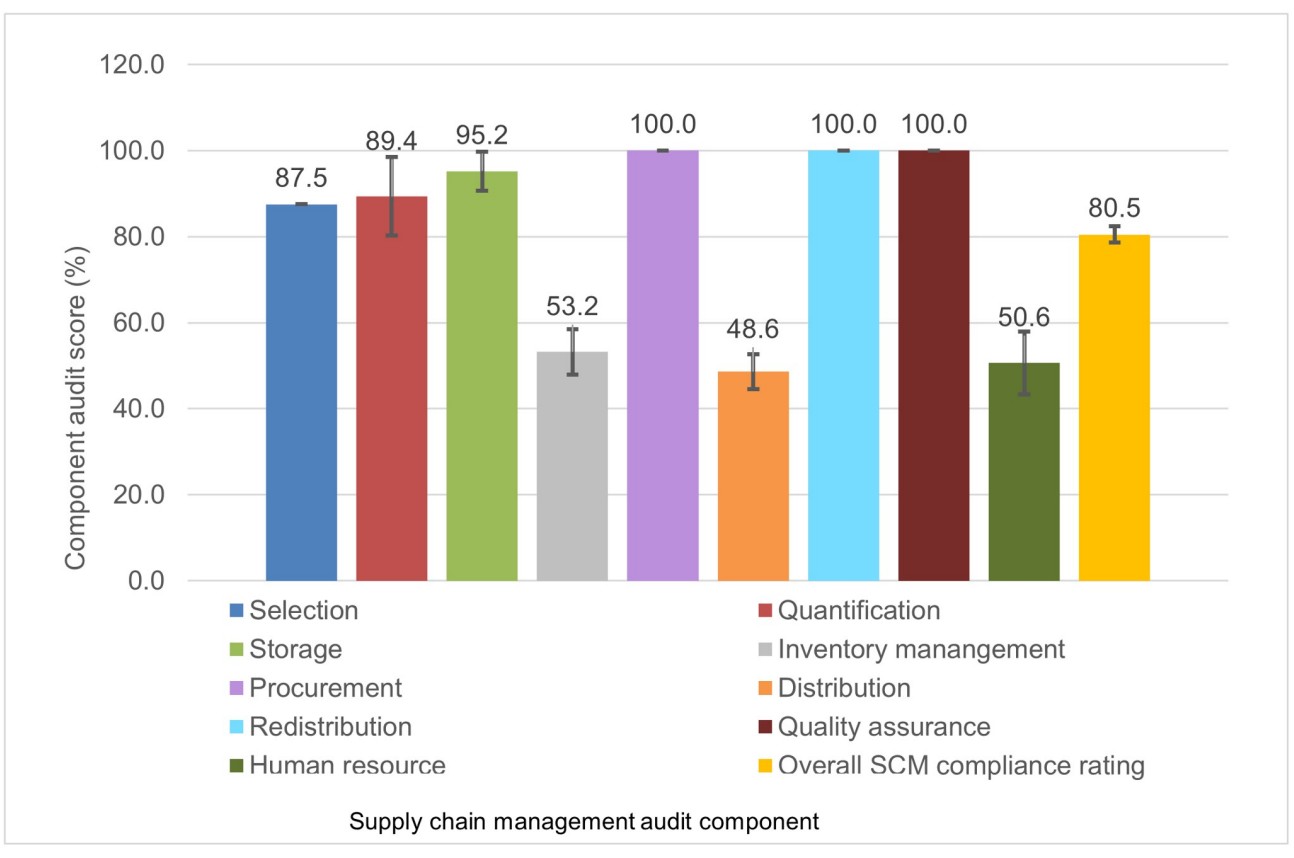

**Fig 3. Average score per supply chain management component of the audited PHC clinics, Mopani District, Limpopo Province, South Africa.**

POC diagnostic tests were user friendly, simple to perform, required minimal training, enabled rapid testing at first visit, and were robust because they did not require refrigeration. Participants reported that there was no wastage of POC tests.

**Quantification.** The mean score for quantification of POC tests was 89.4% (95% CI: 80.2%–98.5%). PHC clinic operational managers are responsible for predicting the demand of POC tests based on seasonal variations (COVID-19 waves) to ensure that PHC clinics had enough POC tests. OPMs in 42 (89.4%) PHC clinics successfully predicted the demand for POC tests. The remaining five (10.6%) PHC clinics predicted the demand based on the number of people visiting the PHC clinic.

**Storage.** The mean score for storage of POC tests was 95.2% (95% CI: 90.7–99.7). Storage facilities were available in 45 (95.7%) PHC clinics. Thirty (66.6%) PHC clinics stored POC tests in the dispensary with other health commodities while 15 (33.3%) stored the POC tests in a separate room that had been converted into a storeroom. Two (4.3%) PHC clinics did not have dedicated storage facilities, but stored POC tests in testing rooms. Most of the PHC clinics (n = 43, 91.5%), had functional air conditioners and professional nurses monitored the temperature of the dispensary storeroom twice daily using a thermometer. Two PHC clinics (4.3%) had non-functional air conditioners and two (4.3%) PHC clinics had no air conditioners.

**Inventory management.** The compliance score for inventory management was 53.2% (95% CI: 47.9%–58.5%). OPMs and pharmacist assistants usually were responsible for inventory management, but occasionally this responsibility was delegated to professional nurses.

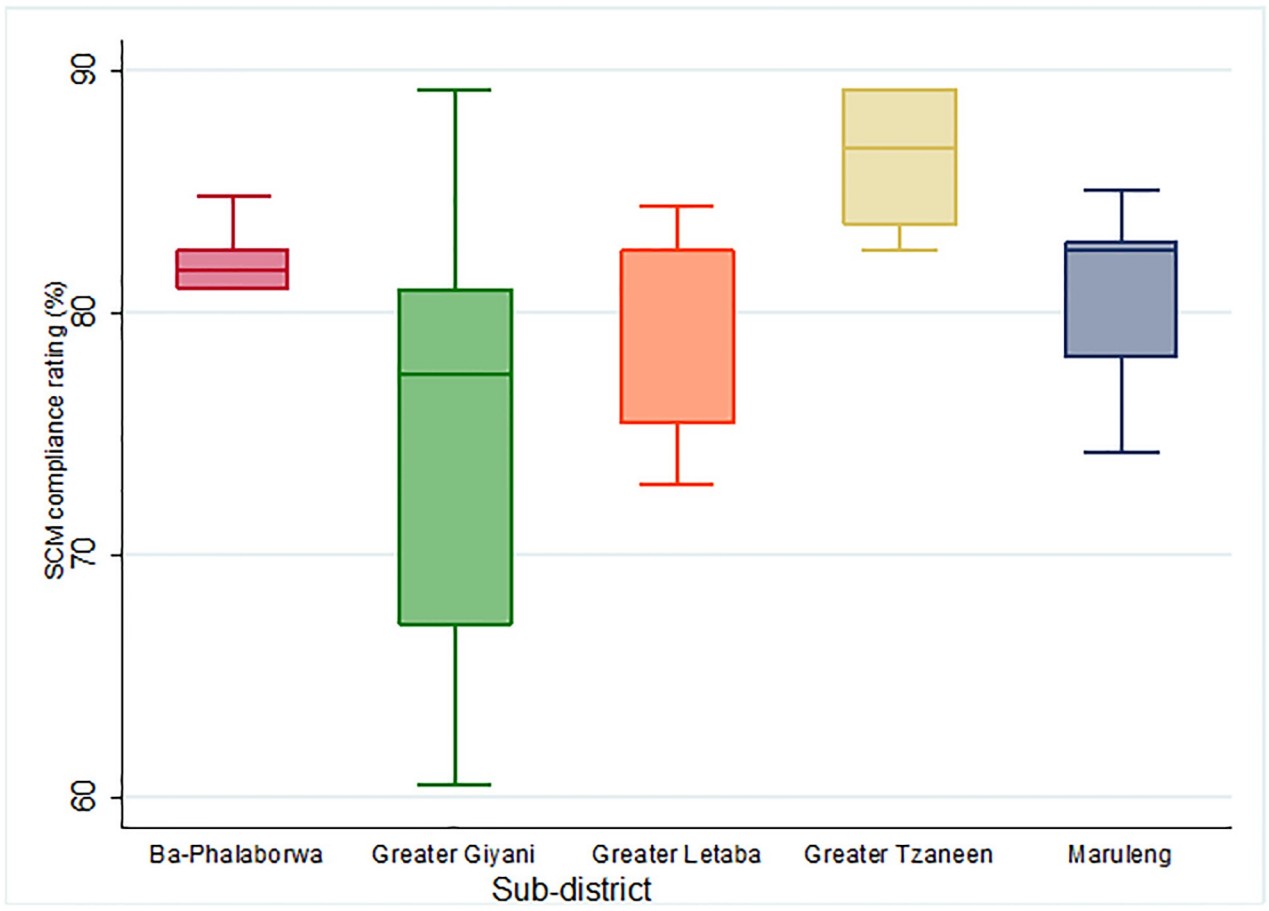

**Fig 4. Variation in the supply chain management audit scores of the audited PHC clinics, in the five sub-districts of Mopani District, Limpopo Province, South Africa.**

Pharmacist assistants did not work on weekends, raising issues of inventory management in their absence. Most PHC clinics (97.9%) had personnel whose duties included managing COVID-19 POC diagnostic tests, while one clinic (2.1%) did not have any inventory management processes in place. An updated list of existing COVID-19 POC tests in the last three months was available in 87.2% of PHC clinics. Most PHC clinics (91.5%) did not have documents for recording the expiry dates of existing COVID-19 POC tests as all the tests were new.

All the PHC clinics used manual systems (stock/bin cards and inventory control forms/books) for inventory management. An updated inventory management list (tests available at the beginning of the week, number of both negative and positive tests conducted, and tests available at the end of the week), was sent to the sub-district managers weekly via WhatsApp by 44 (93.6%) PHC clinics. The sub-district managers then reported to the district office on a weekly basis. Three PHC clinics mentioned that they reported on the National Health Laboratory Service (NHLS) COVID Screening Application (CSA) portal.

**Procurement.** PHC clinics scored 100% for procurement. Requisitions were made from district hospitals and sub-district PHC offices, which were responsible for distributing POC tests. Thirty-four (72.3%) PHC clinics made requisitions based on demand, seven (14.9%) made weekly requisitions, and six (12.8%) made quarterly requisitions. In 32 (68.1%) PHC clinics requisitions were made when one box, containing 25 tests was remaining. In 15 (31.9%)

PHC clinics requisitions were made when 5–10 POC tests were remaining. Ba-Phalaborwa and Maruleng sub-districts reported that they had to submit proof that they had used previously supplied POC tests hence inventory management forms were available in these sub-districts. In all PHC clinics, the turnaround time ranged from daily to weekly. For example, Ba-Phalaborwa sub-district received stock daily because it had a delivery vehicle, a designated driver, and was close to the dispensing sub-district PHC clinic. PHC clinics in sub-districts which made requisitions from district hospitals received their stock on a weekly basis with the delivery of other health commodities. PHC clinics that had mobile clinics and local area drivers collected POC tests from district hospitals. When the requisition was urgent, PHC clinic OPMs would collect directly from district hospitals.

**Distribution.** Distribution had a SCM compliance score of 48.6% (95% CI: 44.6%–52.7%). District hospitals were responsible for distributing COVID-19 POC tests after receiving the requisitions made by the PHC clinics. All 47 PHC clinics reported that they always received stock after making a requisition. No stock outs were reported. Most PHC clinics, 43 (91.5%) reported that they check the delivery note that came with the supplies against the requisition made however there was no system in place to document the differences. Forty-six (97.9%) PHC clinics reported filing all delivery forms in a safe place. Only nine (19.1%) PHC clinics ensured that the driver or delivery person signed the delivery form and only 23 (48.9%) PHC clinics wrote down delivery information in a ledger book.

**Redistribution.** SCM compliance scores for redistribution were 100%. All 47 PHC clinics reported having procedures in place to redistribute COVID-19 POC tests to other facilities when the expiry date was close. All the sub-districts reported having WhatsApp groups for communicating between PHC clinics. Stock rotation between PHC clinics was done through WhatsApp or by completing a short-dated form to return the stock to a district hospital which then rotated the stock to PHC clinics in need. PHC clinics also reported using the first-in, first-out (FIFO) principle. This ensured that diagnostic tests did not expire before being used.

**Quality assurance.** The audit revealed that all PHC clinics had efficient quality assurance measures in place (100% SCM compliance score). All PHC clinics indicated that the person receiving the new batch of COVID-19 test kits verified that the box was properly sealed and that the individual test kits were also sealed upon delivery. All professional nurses also verified that the test kit was sealed before testing.

**Human resource capacity.** The average SCM compliance score for human resource capacity was 50.6% (95% CI: 43.3%–58.0%). As professional nurses were responsible for performing the tests at PHC clinics, training was essential. Provincial and district offices were responsible for training the professional nurses. When first introduced, 44 of the 47 PHC clinics (93.6%) reported that they received training on the POC diagnostic test. Two or three professional nurses per PHC clinic attended the training course, then trained the others at their PHC clinics. The training workshops were facilitated by the district office, either at the PHC clinics or at the district hospitals. Two PHC clinics reported that they received training from officials from Anova Health Institute. Three out of the 47 PHC clinics (6.4%) reported that they were not trained and that they trained themselves. They stated that the test was easy to perform and was similar to other POC tests used to test for diseases such as HIV and diabetes. All PHC clinics reported that training updates would be necessary if a new test were to be introduced, or if the current POC was modified. All the PHC clinics did not have standard operating procedures (SOPs) for performing the COVID-19 test, inventory management, and safe disposal of COVID-19 test kits but most participants reported that they followed COVID-19 test kit instructions. Nine of the 47 PHC clinics (19.1%) reported that they used SOPs for other diseases such as HIV and diabetes.

### Relationship between PHC clinic characteristics and SCM compliance

Overall SCM compliance scores were significantly correlated with clinic headcount (r = 0.38 and p = 0.008 (Fig 5). The larger the headcount of the PHC clinic, the higher the SCM compliance score.

SCM compliance scores were also significantly correlated with the ideal clinic score (r = 0.43, p = 0.003) (Fig 6). A high ideal clinic score represents a PHC clinic with good infrastructure, adequate staff, adequate medicine and supplies, and good administrative processes. A high SCM compliance rating means that the SCM processes in place are satisfactory hence adequate accessibility to COVID-19 POC tests.

## Discussion

This study determined to effect of SCM on accessibility of SARS-CoV-2 POC tests through an audit of 47 PHC clinics in Mopani District. The audit results revealed that none of the audited PHC clinics were compliant (score < 90%) with all the criteria stipulated by the audit tool, based on WHO and MSH guidelines. Clinics scored 100% for procurement, quality assurance, and redistribution, obtained moderate scores (87.5%—95.2%) for selection, quantification, and storage, but scored lowest on inventory management (53.2%), distribution (48.6%), and human resource capacity (50.6%). This study confirmed that access to POC diagnostic tests

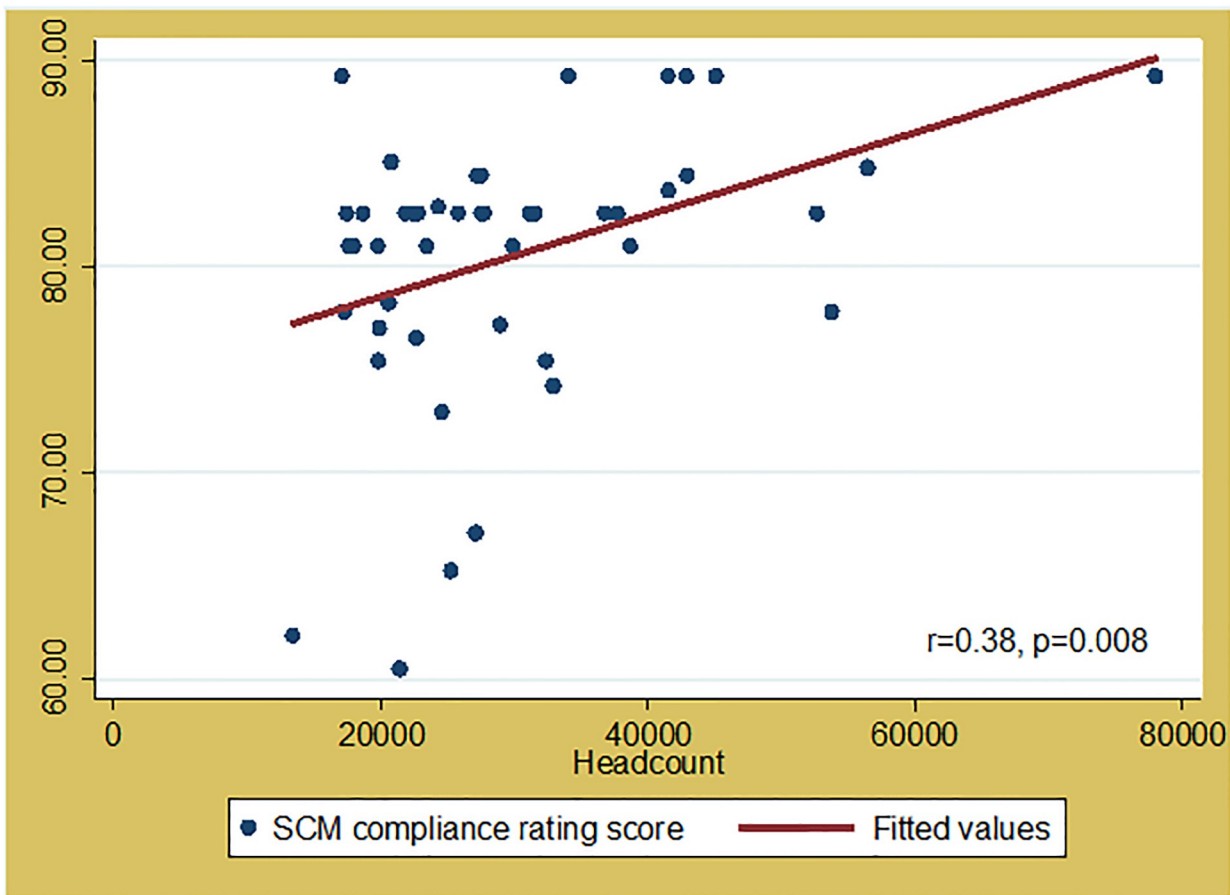

**Fig 5. Correlation between supply chain management compliance score and clinic headcount of the audited PHC clinics in Mopani District, Limpopo Province, South Africa.**

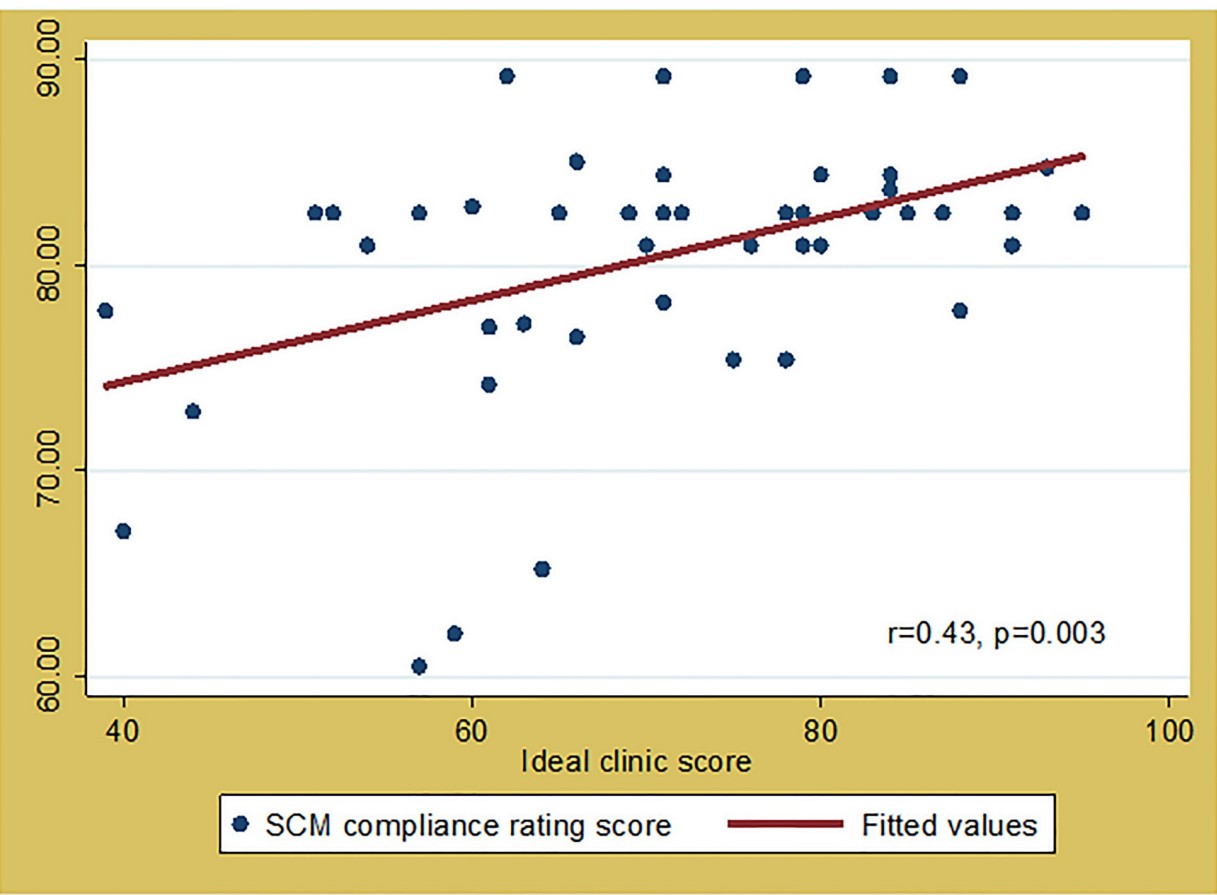

**Fig 6. Correlation between supply chain management compliance score and ideal clinic score of the audited PHC clinics in Mopani District, Limpopo Province, South Africa.**

are impacted by SCM systems especially in resource-limited settings where the distribution of COVID-19 POC tests depends on the availability of POC tests at district hospitals and sub-district offices.

Inventory management, distribution and human resource capacity were found to be the greatest barriers of all the 9 SCM parameters within this study setting. Evidence from other LMICs show that inventory levels usually guide the quantity of stock procured [15, 28] and that adequate inventory levels are facilitated by clear communication between PHC facilities and the provincial or district offices that control the distribution of stock [10, 29, 30]. Currently, in Mopani District other health commodities are managed through the stock visibility system (SVS), a computerized inventory management platform. COVID-19 POC tests are not yet available on SVS hence the manual recording system used at the PHC clinics. Kihara and Ngugi [31] found that good inventory management systems using barcode labels and barcode scanners reduce human error by eliminating manual documentation.

The district hospitals relied on supplies from provincial and national levels. This created a ripple effect. If there is no flow in supply from the national or provincial level to the district hospitals, there will be no flow in supply to the PHC clinics therefore affecting access to POC diagnostic tests. The study found that within this setting, POC tests were usually distributed within 24 hours to a week depending on urgency and availability of delivery vehicles at the district hospitals. This finding agrees with findings by Kuupiel et. al. which revealed that, POC

diagnostics to PHC health facilities in Ghana depended on the availability of the tests at the national and regional medical stores as well as at the district health directorate stores [7].

In the Mopani district, SCM is negatively affected by human resources shortfalls. Nurses are key links in PHC clinic SCM as they are responsible for inventory management, quality assurance, redistribution and procurement. In Uganda, supply chain functions across all levels of care were negatively affected by inadequately skilled personnel, which resulted in staff taking on supply chain functions in addition to their key roles and responsibilities in health facilities, which contributed to poor performance in priority SCM areas [32]. This finding confirmed that adequately trained clinic staff are well empowered to fulfil essential supply chain functions and to make decisions that positively impact health supply availability and supply chain operations [33].

To the best of our knowledge, ours is the first study to evaluate SCM of SARS-CoV-2 POC diagnostic services in resource-limited settings. The 47 PHC clinics audited in this study can be used as a model for other similar resource-limited settings. A strength of our study is that it provides a sound methodology for evaluating SCM in resource limited settings. Limitations include that although the audit tool covered test performance, sensitivity, and specificity, we could only collect data on the perceived sensitivity and specificity of POC tests, without confirmatory laboratory test results. Another limitation was that this audit was only conducted in one district, which doesn't allow comparison with other areas.

### Recommendations

Based on the findings of this study, we recommend the following:

- Adoption of efficient inventory management tools, especially software systems, linking national, provincial, district, and PHC facilities. Currently, inventory is tracked manually, which may result in inefficient logging of orders or deliveries, especially in understaffed environments.

- Implementation of a robust human resource management system by adopting a nurse-centric approach to improve SCM.

- Structured training course for POC SCM in resource-limited settings such as our study area to help improve compliance to standards. Additionally, we recommend regular workshops for PHC clinic staff to improve distribution of POC tests that are inadequately managed due to documentation challenges. SOPs for testing, inventory management, and safe disposal of test kits should be available to staff to prevent them from working blindly.

- Strengthening the supply chain and logistics at a national level to ensure that high quality POC test kits and consumables are available at all testing sites and that stock outs are minimized.

- A follow up study to evaluate SCM of SARS-CoV-2 POC tests at a provincial or national level in order to evaluate the selection and procurement process. This will provide an opportunity to evaluate the SCM at a higher level and establish the barriers and enablers of all 9 components of the SCM system in detail.

### Conclusion

Our results revealed that the PHC clinics in the Mopani District, Limpopo, South Africa do not comply with international SCM guidelines. The audit results revealed deficiencies in inventory management, distribution, and human resource capacity. PHC clinics were

compliant in procurement, redistribution, and quality assurance. The audit also revealed variations in the SCM compliance scores between the sub-districts showing inconsistencies in the SCM processes currently in place. We highly recommend the adoption of centralized online inventory management tools and structured training for healthcare workers on SCM of POC diagnostics. SCM strategies for POC diagnostics need to be well planned to ensure accessibility of POC diagnostic services in rural resource-limited settings.

## Supporting information

**S1 Fig. Supply chain management audit tool.**
(DOCX)

**S1 Table. Characteristics of the 47 participating PHC clinics in Mopani District.**
(DOCX)

## Acknowledgments

The authors would like to thank the authorities of the Mopani District, the sub-district clinic managers, and the operational managers for granting us permission to conduct the study. We thank the clinic staff who participated in this study for their valuable input. The authors would also like to extend their appreciation to Ms. Dorothy Southern & Dr. Cheryl Tosh for editing, Dr Davison Moyo for statistical assistance, the School of Health Systems and Public Health, University of Pretoria for supporting the development of this research study, and the National Research Foundation & Ninety-One SA (Pty) Ltd for the financial assistance through scholarships granted to the PI.

## Author Contributions

**Conceptualization:** Kuhlula Maluleke, Tivani Mashamba-Thompson.

**Data curation:** Kuhlula Maluleke.

**Formal analysis:** Kuhlula Maluleke, Alfred Musekiwa.

**Methodology:** Kuhlula Maluleke, Tivani Mashamba-Thompson.

**Supervision:** Tivani Mashamba-Thompson.

**Writing – original draft:** Kuhlula Maluleke.

**Writing – review & editing:** Alfred Musekiwa, Tivani Mashamba-Thompson.

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
