## [Decision Letter · Decision Letter 0]

7 Feb 2023

PONE-D-22-33471Evaluating supply chain management of SARS-CoV-2 point-of-care (POC) diagnostic services in resource-limited settingsPLOS ONE

Dear Dr. Maluleke,

Thank you for submitting your manuscript to PLOS ONE. After careful consideration, we feel that it has merit but does not fully meet PLOS ONE’s publication criteria as it currently stands. Therefore, we invite you to submit a revised version of the manuscript that addresses the points raised during the review process.

We look forward to receiving your revised manuscript.

Kind regards,

Mathumalar Loganathan Fahrni

Academic Editor

PLOS ONE

Journal Requirements:

2. We note that Figure 1 in your submission contain map image which may be copyrighted. All PLOS content is published under the Creative Commons Attribution License (CC BY 4.0), which means that the manuscript, images, and Supporting Information files will be freely available online, and any third party is permitted to access, download, copy, distribute, and use these materials in any way, even commercially, with proper attribution. For these reasons, we cannot publish previously copyrighted maps or satellite images created using proprietary data, such as Google software (Google Maps, Street View, and Earth). For more information, see our copyright guidelines: http://journals.plos.org/plosone/s/licenses-and-copyright.

Additional Editor Comments:

See attachment

Reviewers' comments:

Reviewer's Responses to Questions

**Comments to the Author**

1. Is the manuscript technically sound, and do the data support the conclusions?

Reviewer #1: Yes

Reviewer #2: Partly

2. Has the statistical analysis been performed appropriately and rigorously? 

Reviewer #1: Yes

Reviewer #2: Yes

3. Have the authors made all data underlying the findings in their manuscript fully available?

Reviewer #1: Yes

Reviewer #2: Yes

4. Is the manuscript presented in an intelligible fashion and written in standard English?

Reviewer #1: Yes

Reviewer #2: No

5. Review Comments to the Author

Reviewer #1: This is an important topic in diagnostics as access is usually impacted by supply chain parameters. A couple of comments needs to be addressed.

1. The abstract should state that the SCM scoring was based on a questionnaire that was developed by the authors. Otherwise it gives an impression that it is some international scoring mechanism

2. The conclusion section is not strong enough. It is important to mention why there is a lot of variation in the districts in the audit parameters?

3. It should be mentioned that the limitation of this study is that only one area was surveyed. Whether these SCM issues are related to this area (as it is a rural area) or it is a bigger issue, we still do not know.

Reviewer #2: The authors surveyed the supply chain management (SCM) of SARS-CoV-2 point-of-care (POC) diagnostic services in selected 47 clinics in South Africa. The authors found that they did not comply with international SCM guidelines.

The clinical significance of this survey is limited, and I would recommend rejection. I suggest that the authors “evaluate” the influence on the quality of POCT diagnostic services between different levels of storage, selection, quantification, inventory management, distribution, and human resource capacity.

6. PLOS authors have the option to publish the peer review history of their article (what does this mean?). If published, this will include your full peer review and any attached files.

Reviewer #1: No

Reviewer #2: No

---

## [Author Response · Author response to Decision Letter 0]

19 Mar 2023

A rebuttal letter that addresses all the reviewers comments has been attached

---

## [Decision Letter · Decision Letter 1]

12 Apr 2023

PONE-D-22-33471R1Evaluating supply chain management of SARS-CoV-2 point-of-care (POC) diagnostic services in primary healthcare clinics in Mopani District, Limpopo Province, South AfricaPLOS ONE

Dear Dr. Maluleke,

Thank you for submitting your manuscript to PLOS ONE. After careful consideration, we feel that it has merit but does not fully meet PLOS ONE’s publication criteria as it currently stands. Therefore, we invite you to submit a revised version of the manuscript that addresses the points raised during the review process.

We look forward to receiving your revised manuscript.

Kind regards,

Hamufare Dumisani Dumisani Mugauri, Ph.D. Public Health

Academic Editor

PLOS ONE

Reviewers' comments:

Reviewer's Responses to Questions

**Comments to the Author**

1. If the authors have adequately addressed your comments raised in a previous round of review and you feel that this manuscript is now acceptable for publication, you may indicate that here to bypass the “Comments to the Author” section, enter your conflict of interest statement in the “Confidential to Editor” section, and submit your "Accept" recommendation.

Reviewer #1: All comments have been addressed

Reviewer #2: (No Response)

2. Is the manuscript technically sound, and do the data support the conclusions?

Reviewer #1: Partly

Reviewer #2: No

3. Has the statistical analysis been performed appropriately and rigorously? 

Reviewer #1: Yes

Reviewer #2: N/A

4. Have the authors made all data underlying the findings in their manuscript fully available?

Reviewer #1: Yes

Reviewer #2: Yes

5. Is the manuscript presented in an intelligible fashion and written in standard English?

Reviewer #1: Yes

Reviewer #2: Yes

6. Review Comments to the Author

Reviewer #1: (No Response)

Reviewer #2: I felt that majorities of my previous comments remained unaddressed.

I felt that majorities of my previous comments remained unaddressed.

7. PLOS authors have the option to publish the peer review history of their article (what does this mean?). If published, this will include your full peer review and any attached files.

Reviewer #1: No

Reviewer #2: No

---

## [Author Response · Author response to Decision Letter 1]

13 May 2023

I am submitting this revised manuscript in response to the comments from Reviewer 2. The reviewer suggested that we evaluate the influence of different SCM factors, such as storage, selection, quantification, inventory management, distribution, and human resource capacity, on the quality of POCT diagnostic services. However, this was beyond the scope of our project, which aimed to evaluate the SCM for SARS-CoV-2 POC diagnostic services in resource-limited settings and identify barriers and enablers of accessibility to SARS-CoV-2 diagnostic services. We used an audit tool that was guided by relevant guidelines, but we focused only on aspects pertaining to SCM.

The reviewer also suggested that we use an appropriate format for reporting a clinical audit. However, we would like to clarify that this was not a clinical audit, but an audit evaluating the SCM of the POC diagnostic test.

We attempted to reach out to the reviewer twice for further suggestions on how to address their concerns, but we did not receive any response. Therefore, we have resubmitted the article without further clarification due to time constraints on our side. The initial submission was on December 9th, 2022.

We hope that this revised manuscript meets the requirements for publication.

---

## [Editor Report · Decision Letter 2]

6 Jun 2023

Evaluating supply chain management of SARS-CoV-2 point-of-care (POC) diagnostic services in primary healthcare clinics in Mopani District, Limpopo Province, South Africa

PONE-D-22-33471R2

Dear Dr. Maluleke,

We’re pleased to inform you that your manuscript has been judged scientifically suitable for publication and will be formally accepted for publication once it meets all outstanding technical requirements.

Kind regards,

Hamufare Dumisani Dumisani Mugauri, Ph.D. Public Health

Academic Editor

PLOS ONE

---

## [Editor Report · Acceptance letter]

19 Jun 2023

PONE-D-22-33471R2 

Evaluating supply chain management of SARS-CoV-2 point-of-care (POC) diagnostic services in primary healthcare clinics in Mopani District, Limpopo Province, South Africa 

Dear Dr. Maluleke:

I'm pleased to inform you that your manuscript has been deemed suitable for publication in PLOS ONE. Congratulations! Your manuscript is now with our production department. 

Kind regards, 

on behalf of

Mr Hamufare Dumisani Dumisani Mugauri 

Academic Editor

PLOS ONE